# Assessment of Restoration Effects and Invasive Potential Based on Vegetation Dynamics of Pitch Pine (*Pinus rigida* Mill.) Plantation in Korea

**Hansol Lee [1], Ji Hong An [2], Hyun Chul Shin [3] and Chang Seok Lee [4],***

[1]    Department of Biology, Miami University, Oxford, OH 45056, USA; leeh28@miamioh.edu
[2]    Baekdudaegan National Arboretum, Bonghwa 36209, Korea; jhan@bdna.or.kr
[3]    National Institute of Ecology, Seocheon 33657, Korea; othello1@hanmail.net
[4]    Department of Bio and Environmental Technology, Seoul Women's University, Seoul 01797, Korea
*    Correspondence: leecs@swu.ac.kr; Tel.: +82-2-970-5666

**Abstract:** During the period of Japanese occupation (1910–1945) and the Korean War (1950–1953), extensive areas of forest were severely degraded by over-harvesting and fire in Korea. In addition, intensive use of the forest-resources to obtain fuel, organic compost, livestock feed, and so on contributed to forest degradation. As a result, the South Korean government launched large-scale tree planting projects to reforest the denuded mountains particularly in the 1960s. This study aims to evaluate the restoration effects of the pitch pine (*Pinus rigida* Mill.) plantations and further diagnose the invasive potential of the pitch pine. To arrive at the goals, we investigated the changes of vegetation and soil characteristics in different chronosequences in the pitch pine plantations and in native forests, which were selected as reference stands. Pitch pine plantations were usually planted on mountainous land, which is characterized by an elevation of below 300 m above sea level and a gentle slope below 20°. The species composition of the pitch pine forestations was different depending on the study site but tended to resemble that of the reference stands in the years after forestation. The species diversity showed an increasing trend in response to stand age. The frequency distribution of diameter classes of dominant tree species showed a trend for pitch pine plantations to succeed to native oak stands. A change in canopy profiles depending on stand age also proved the successional trend. The establishment and development of pitch pine plantations for reforestation contributed to erosion control and improved the physic-chemical properties of the soil and thus prepared a basis for the recovery of native vegetation. Such changes in vegetation and soil confirmed that the pitch pine plantations successfully achieved the restoration goals. On the other hand, mature pitch pine stands reproduced young pitch pine stands by self-seeding on the slopes of various sorts of roads including expressways. This shows that pitch pine is successfully established in Korea and thereby the species has been naturalized. However, the natural succession of pitch pine stands in Korea suggests that it is possible to introduce some exotic species for reforestation without resulting in uncontrolled invasion.

**Keywords:** deforestation; exotic species; Korea; pitch pine; plantation; restoration; succession

---

## 1. Introduction

Most plantations have a pulp, fuelwood or timber production function but they can also be established for soil and water conservation, for the restoration of abandoned pastures, to provide shade for crops, for carbon sequestration or for a combination of multiple objectives such as timber production or carbon mitigation, and poverty reduction [1]. In fact, plantations can function as habitat for a wide range of forest organisms [2–5]. They can act as instant forest ecosystems in which the regeneration of

trees and succession is often facilitated by favorable understory microclimatic conditions, increased vegetation structural complexity, suppression of grasses and the development of a litter and humus layer [6,7].

In Korea, the intention seems largely to correspond to the latter, as most forestation projects have applied for erosion protection. In this respect, the goals of forestation also include the concept of nature restoration or preventing further damage. This concept corresponds with the objectives that the restoration project pursues today, such as sustaining the natural balance and improving environmental conditions through the function of the restored ecosystem [8,9].

The introduction of pitch pine (*Pinus rigida* Mill.) to Korea dates back to the 19th century. However, the South Korean government launched large-scale tree planting campaigns to reforest denuded mountains after deforestation during the Japanese occupation period (1910–1945) and the Korean War (1950–1953) particularly in the 1960's [10–12]. During the period of Japanese occupation (1910–1945) and the Korean War (1950–1953), a wide range of forest in Korea was rigorously degraded by over-cutting and fire. The deforestation of South Korea during Japanese occupation (1910–1945) and Korean War (1950–1953) was very serious, so much so that 42% of the total forest area (28,000 km$^2$) of the country was stripped of trees. Moreover, 19% of the denuded area (5200 km$^2$) needed urgent forestation to prevent erosion. The average stand volume declined from 100 cubic m ha$^{-1}$ to 10.6 cubic m ha$^{-1}$ at that time [13]. The litter layer and the development of the A-layer of the forest floor became very thin, resulting in relatively low organic matter and nutrients contents in the soil and consequently frequent erosion occurred [14]. Since deforestation was so extensive, reforestation was made a national priority. In the 1960s and 1970s, the South Korean government performed large-scale reforestation by introducing fast growing trees (i.e., pitch pine, Japanese larch, *Larix leptolepic* S. et Z.) and fertilizing trees (i.e., black locust, *Robinia pseudoacacia* L. and alders, *Alnus* spp.) to restore the severely degraded forest [4,5,13,15]. Pitch pine was considered a promising tree for reforestation due to its fast growth and great adaptability [14]. Consequently, pitch pine became one of the most extensively planted exotic plants for forest restoration in South Korea [16]. However, its potential for aggressive invasiveness was overlooked during these reforestation campaigns [17,18].

The process of deforestation results in fragmentation of the remaining patches [19], loss of biodiversity with a particular concern for endangered species [20,21], carbon emissions to the atmosphere [22], and degradation of the natural resources, mainly soil and water [23,24]. Forest restoration restores natural vegetation in disturbed areas and improves the health of a stand [25]. Forest restoration has received increasing attention from governments and ecologists since the results of deforestation have been made evident, i.e., serious environmental issues and the loss of ecological services [2,26–28]. Restoration ecology aims to assist and to manage a natural ecosystem, degraded by human-induced activities, to recover its ecological integrity to a maximum level of biodiversity and variability of the ecological processes [29–31]. Current restoration approaches have focused on the recovery of composition, structure, natural habitat, ecosystem processes and services [2,26,28,32–35]. If it is to be successful in conserving biodiversity, restoration must go beyond the reconstruction of structure, composition, and the appearance of a site [36,37] and restore biological interactions, processes, and integrity [38–43].

Restoration success can be viewed as a continuum from the successful establishment of the initial planting to the successful establishment of attributes that ensure a self-sustaining, functioning natural system. Although the later stages of this continuum are the most likely goals for restoration projects, the initial stages must be successful if the longer term goals are to be met [44]. Through success in the initial plantings, restoration becomes a tool to conserve biodiversity as new individuals colonize the restored habitat. Following the successful establishment of those conditions required for recolonization and establishment, individuals and species become a part of and contribute to the maintenance of the system. Evaluation of restoration success can be achieved by describing restoration success in the context of a continuum. If conditions are suitable for the first stage of restoration (species colonization and establishment), then it is likely that the restoration will continue to a later stage [45,46].

Plant invasion is a global issue that can potentially threaten biodiversity and also human health; it also has large economic costs [47,48]. However, scientists understanding of the factors controlling invasive plant distribution and abundance is limited [49–53]. Whether introduced plants, either intentionally or accidentally, become invasive is an important but little understood issue in the study of how such plants affect native communities and local ecosystems. Plantations may provide an important avenue for invasion of alien plant species [54–56].

The widespread invasion of introduced pine species has been found in many places around the world, providing a useful natural experiment for determining the factors that affect the degree to which pines can invade different habitats and vegetation types [17,18,57]. There are several cases of other pine invasions worldwide [17,18,53,54,57]. However, studies about pitch pine invasion are rare [58] and no study has been done on their introduction, naturalization, and invasion in South Korea. Therefore, this study aims to contribute to the understanding of the adaptation of pitch pine as an exotic plant.

In the present study, we investigated the vegetation and soil characteristics of pitch pine plantations in different successional stages and natural forests selected as reference stands. The objectives of the current study were to: (1) evaluate the restoration effects of pitch pine afforestation based on changes in vegetation and soil properties in the pitch pine plantations with different ages and a comparison of the results to those of the reference oak forests; (2) diagnose the invasive potential of pitch pine based on the expansion and decline of the plant depending on the habitat condition.

## 2. Materials and Methods

### 2.1. Study Areas

Sites for this study were selected in stands with diverse ages. Mts. Chilbo, Nam, Bulam, Baekwoon, Sori, Suri, and Deokyu (hereafter abbreviated as Mts. Cb, Na, Ba, Bw, Sb, Sr, and Dy, respectively), which were afforested with pitch pine (*Pinus rigida* Mill.) 33, 46, 48, 52, 53, 56, and 86 years ago, respectively, were chosen as study sites (Figure 1, Table 1). Native oak (*Quercus mongolica* Fisch. Ex Ledeb.) stands, which are located near to each study sites and established naturally, were chosen as the reference plots. Mongolian oak forest is considered to be the first-class in the ecological naturalness as the representative forest of the late successional stage in Korea [59].

The sites selected for this study have similar soil, brown forest soil developed on granite or metamorphic rock; climate, a temperate deciduous forest zone; and topography, below mid-slope. The climate is continental, with warm and moist summers, and cold and dry winters (Table 1) [60–62]. The mean annual temperature ranged from 10.2 °C (Mt. Sb) to 12.8 °C (Mt. Ba) and the mean annual precipitation ranged from 1126.3 mm (Mt. Dy) to 1441.3 mm (Mt. Ba) (Table 1).

A 30 year old pitch plantation is composed of three layers: canopy tree, shrub, and herb, without an understory tree layer. *Quercus mongolica* Fisch. Ex Ledeb., *Rhododendron mucronulatum* Turcz., and *Symplocos chinensia* for. *pilosa* (Nakai) Ohwi., etc., and *Spodiopogon sibiricus* Trin., *Carex humilis* var. *nana* (H. Lév. and Vaniot) Ohwi, *Pyrola japonica* Klenze ex Alef., etc., dominate the shrub and herb layers, respectively. Pitch pine plantations of the older stages are composed of four layers: canopy tree, understory tree, shrub, and herb. The understory tree layer of the older plantations is dominated by *Quercus* spp. including *Q. mongolica*. However, there is little difference in the species composition of the shrub and herb layers. In 40 year old pitch plantations, *R. mucronulatum*, *Rhododendron schlippenbachii* Max., *Callicarpa japonica* Thun., etc., and *S. sibiricus* Trin., *C. humilis* var. *nana* (H. Lév. and Vaniot) Ohwi, *Artemisia keiskeana* Miq., etc., dominate the shrub and herb layers, respectively. Shrub and herb layers of 50 year old pitch plantations are dominated by *R. mucronulatum*, *Styrax japonicas* Siebold and Zucc., *Viburnum erosum* Thunb., etc., and *S. sibiricus* Trin., and *Carex humilis* var. *nana* (H. Lév. and Vaniot) Ohwi, *Pyrola japonica*, etc., respectively. In 80 year old pitch plantations, *Lindera obtusiloba* Blume, *Lespedeza maximowiczii* C.K. Schneid., *Rhus trichocarpa* Miq., etc., and *S. sibiricus* Trin., *Isodon inflexus* (Thunb.) Kuda, *Atractylodes japonica* Koidz. Ex Kitam., etc., dominate the shrub and herb layers, respectively. The Mongolian oak forest designated as the reference forest is composed of four

layers: the canopy tree, understory tree, shrub, and herb layers, which are dominated by *Quercus mongolica*, *Acer pseudosieboldiana* (Pax) Kom., *Lindera obtusiloba*, *Cornus kousa* F. Buerger ex Miquel, etc., *R. schlippenbachii* Max., *Symplocos chinensia* for. *pilosa* (Nakai) Ohwi., *Callicarpa japonica* Thun., etc., *Ainsliaea acerifolia* Sch. Bip., *Carex siderosticta* Hance, *Disporum smilacinum* A. Gray, respectively.

On the other hand, a young (nine-year-old) pitch pine stand, which was established naturally on the incised slope bordered on an expressway from seeds dispersed from the surrounding mature pitch pine stand (47 years old) was chosen to investigate naturalization and the invasive potential of pitch pine. Self-seeded young pitch stands of this type cover both sides of the expressway for about 60 km; the study site was selected in a representative place among them. This site is located in Cheongyang, central western Korea (Figure 1). This site also has similar environmental conditions to the other study sites, but the gradient was different as it was artificially created. The mean annual temperature was 11.5 °C and the mean annual precipitation was 1331.7 mm [60]. The soil was brown forest soil developed on metamorphic rock [61]. The topographical conditions were as follows: a northwest slope, with an elevation of 80 m, and a gradient of up to 35°.

Pitch pine plantations cover 3411.27 km$^2$, accounting for more than 5% of the total forest area in South Korea [62]. Pitch pine was usually planted in the western parts of South Korea, which has a low elevation and gentle slope compared with the eastern parts and thus the forest there is degraded due to excessive use (Figure 2). The plantations were concentrated below 300 m above sea level and on 20°. However, the plantations did not show any difference depending on aspects (Figure 3).

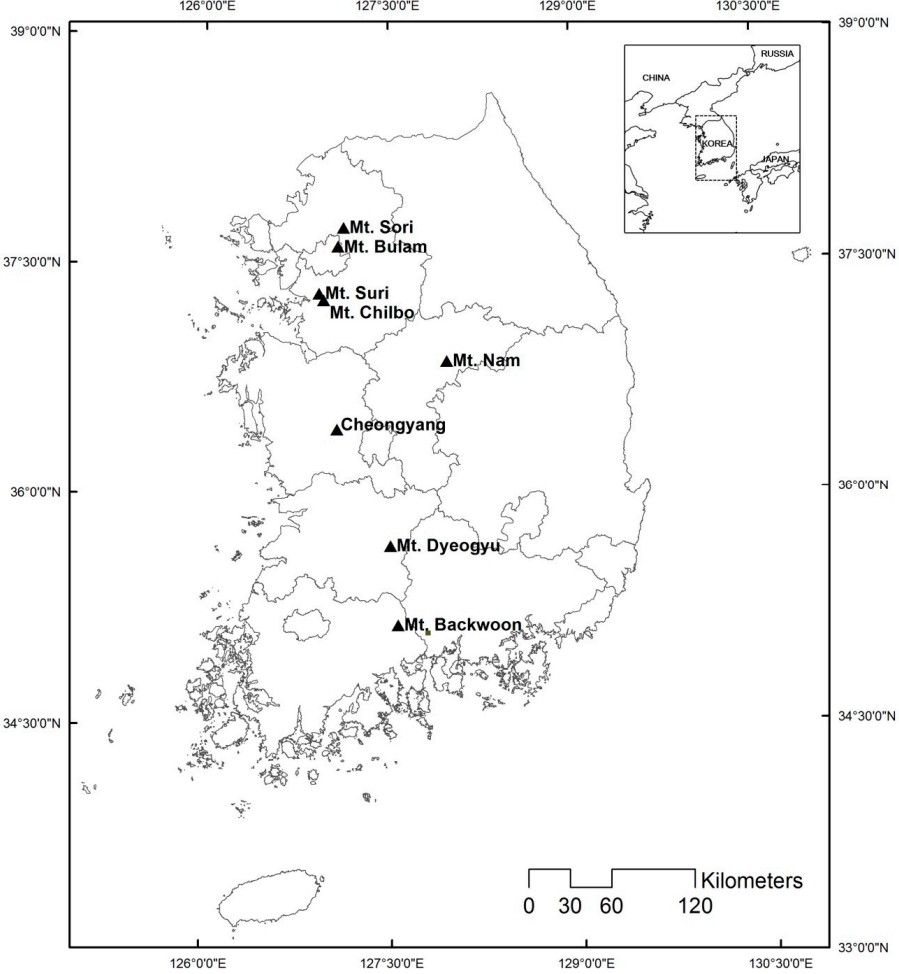

**Figure 1.** A map showing the study sites.

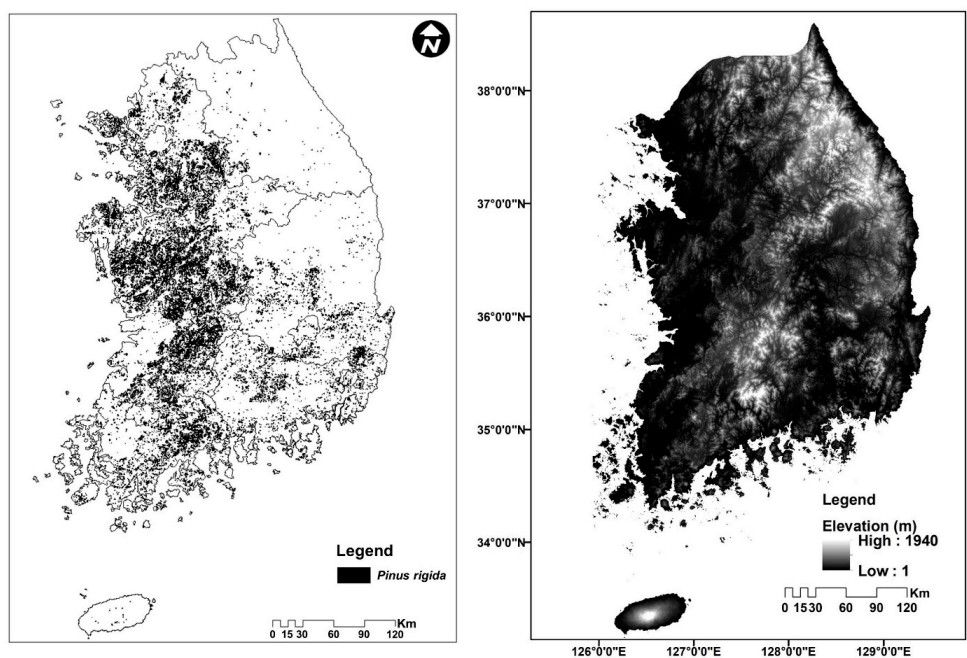

**Figure 2.** Maps showing the nationwide distribution of pitch pine plantations (**left**) and topographic conditions (**right**) in South Korea.

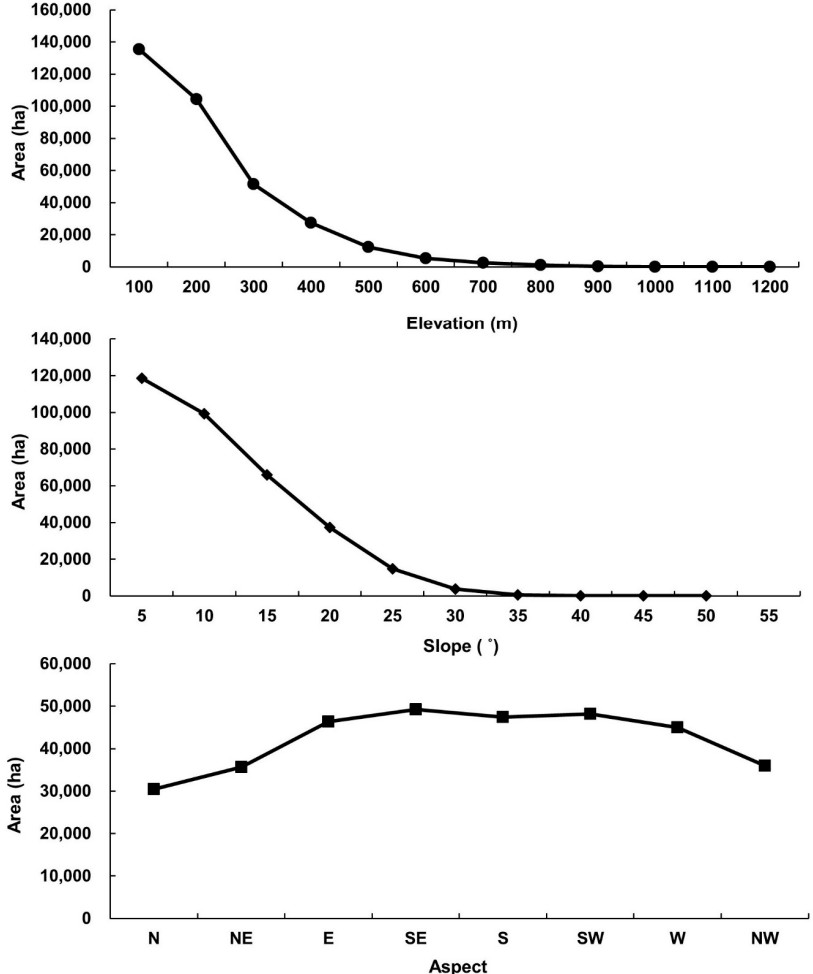

**Figure 3.** Spatial distribution of pitch pine plantations depending on elevation, slope degree, and aspect.

**Table 1.** Stand age and environmental conditions of pitch pine plantations studied.

| Study Site | Stand Age (Years) | Aspect | Elevation (m) | Slope (°) | Mean Temperature (°C) | Mean Precipitation (mm) | Parent Rock | Soil |
|---|---|---|---|---|---|---|---|---|
| Mt. Chilbo | 33 | SW | 65 | 8 | 12.5 | 1330.2 | Granite | Brown forest |
| Mt. Nam | 46 | SE | 161 | 10 | 11.7 | 1235.5 | Metamorphic | Brown forest |
| Mt. Bulam | 48 | NW | 34 | 15 | 12.8 | 1441.3 | Granite | Brown forest |
| Mt. Backwoon | 52 | SE | 154 | 20 | 13.4 | 1341.8 | Granite | Brown forest |
| Mt. Soribong | 53 | NE | 154 | 20 | 10.2 | 1364.8 | Metamorphic | Brown forest |
| Mt. Suri | 56 | S | 56 | 15 | 12.5 | 1330.2 | Metamorphic | Brown forest |
| Mt. Deogyu | 86 | S | 713 | 20 | 11.6 | 1126.3 | Granite | Brown forest |

### 2.2. Experimental Design

This study was carried out to evaluate the effect of ecological restoration resulting from the afforestation of pitch pine by analyzing the changes in vegetation and soil in pitch plantations with different stand ages and by comparing the results with those of natural forests in the area. The 30-, 40-, 50- and 80-year-old pitch pine plantations were selected for the study. The reference forest, used for comparison, was designated as the Mongolian oak forest, which is the representative late successional vegetation type in Korea. The reference forest was selected in a site close to the experimental site (the pitch plantation). The reference forest was selected as a forest, which is from 50 to 100 years old that can form a stable forest and avoid the old growth forest. The sites for this study were selected in the areas with similar climate, soil, and topographic conditions to avoid variation depending on place.

Changes in vegetation were analyzed based on species composition, species diversity, canopy profile, and diameter distribution of the dominant species. Changes in soil were analyzed based on changes in the physic-chemical properties.

Meanwhile, a self-seeded young pitch stand was selected to diagnose the naturalization and invasive potential of the pitch pine.

### 2.3. Mapping and Analysis

A nationwide distribution map was prepared by analyzing the physiognomic map that the Korea Forest Service provides. Construction and analysis of the map were executed using ArcGIS 9.0 and Arcview 3.3 [63]. The isopleth map of elevation was extracted from the digital topographic map on the scale 1:25,000, and a 30 m × 30 m digitalized elevation model (DEM) was prepared on the map. Grid maps of elevation, slope, and aspect were based on the DEM. The zonal statistics function of ArcGIS 9.0 spatial analyst was used to analyze the topographical characteristics of the pitch pine plantation.

### 2.4. Vegetation Analysis

Field surveys of the pitch pine plantation were carried out in 78 plots sized of 20 m × 20 m from June to August 2014. The number of plots chosen for the vegetation survey in each site was 15, 13, 7, 15, 8, 16, and 4 in Mts. Cb, Na, Ba, Bw, Sb, Sr, and Dy, respectively. Surveys of the native oak forests selected as the reference for comparison were carried out in 49 plots by seven plots for each study site. Vegetation surveys were carried out by recording cover using the Braun-Blanquet [64] scale. All plant species appearing in each plot were identified, following Lee [65], Park [66] and the Korean Plant Names Index [67]. Dominance of each species in each plot was estimated using the ordinal scale of cover class based on the Braun-Blanquet [64] scale, and each ordinal scale was converted into the median value of percent cover range in each cover class. The importance value of each species was then calculated by the sum of the relative coverage [68]. Relative coverage was determined by multiplying 100 to the fraction of each species cover to the summed cover of all species in each plot. Finally, a matrix of importance values for all species in all plots was constructed. For the matrix of importance values, relativization by the species column was performed, and it was fed into the Detrended Correspondence Analysis (DCA) for ordination [69] using PC-ORD 5.0 [70].

To describe and compare species diversity and dominance among the study areas, species rank–abundance curves [32,71,72] were plotted. The data used to obtain the species rank–dominance curves were collected from four plots (20 m × 20 m) in all study areas.

The canopy profile was prepared by measuring the coverage and thickness of the canopy in each layer of the vegetation composed of canopy tree, understory tree, shrub, and herb layers within the study plots. The height was measured by hypsometer meter (Haglof, Vertex IV-BT-360). The thickness of the canopy was calculated from the difference between the highest and the lowest height of the canopy forming each layer. Coverage was estimated using the Domin-Krajina scale [73]. The stand profile was prepared by carefully depicting the horizontal and vertical distribution of the major plant species appearing in a belt transect installed with a width of 10 m.

Stem diameters (at breast height for mature trees or at stem base for seedlings and saplings below 1.3 m in height) of tree species were measured and sorted by diameter classes. In order to predict the successional trends of each stand, frequency distribution diagrams according to diameter class of the tree species were prepared [15,74,75].

### 2.5. Soil Analysis

The thickness of the litter layer was measured using a stick ruler after the soil profile was investigated using scissors and a shovel. Soil samples were collected in June to August 2014 from the top 10cm after removing the litter at five random points in each plot; after which, they were pooled, air-dried at room temperature, and sieved through 2 mm mesh. Three, nine, six, and three soil samples were collected in the 30-, 40-, 50-, and 80-year-old pitch pine stands, respectively and total seven soil samples by one sample in each site were done in the reference oak stands.

Organic matter content was determined from the ash-free dry weight after ignition in a muffle furnace at 600 °C for 4 h [76]. The acidity (pH) in a 1:5 w/v mixture of soils and deionized water was measured with a bench top probe. Total nitrogen was measured with the micro-Kjeldahl method [77]. Available P was extracted in 1-N ammonium fluoride (pH = 7.0) and exchangeable K, Ca, and Mg were extracted with 1-N ammonium acetate (pH = 7.0) and measured by inductively coupled plasma atomic emission spectrometry (ICP; Shimadzu ICPQ-1000) [78].

### 2.6. Statistical Analysis

Detrended correspondence analysis (DCA) is an eigenvector ordination technique based on correspondence analysis (CA or RA). It is especially suited to the analysis of ecological data sets based on sample units and species [79,80]. The difference in species composition among stands with different stand ages was analyzed using DCA.

## 3. Results

### 3.1. Species Composition

The species composition of pitch pine plantations showed different site specificity. The species composition of pitch pine plantations is not only becoming similar to each other, but also resembled that of the reference oak stands over stand age (excepting the 86-year-old stands located on high-altitude valleys, which differ from the other pitch pine stands established on the mountain slopes at low altitudes) (Figure 4).

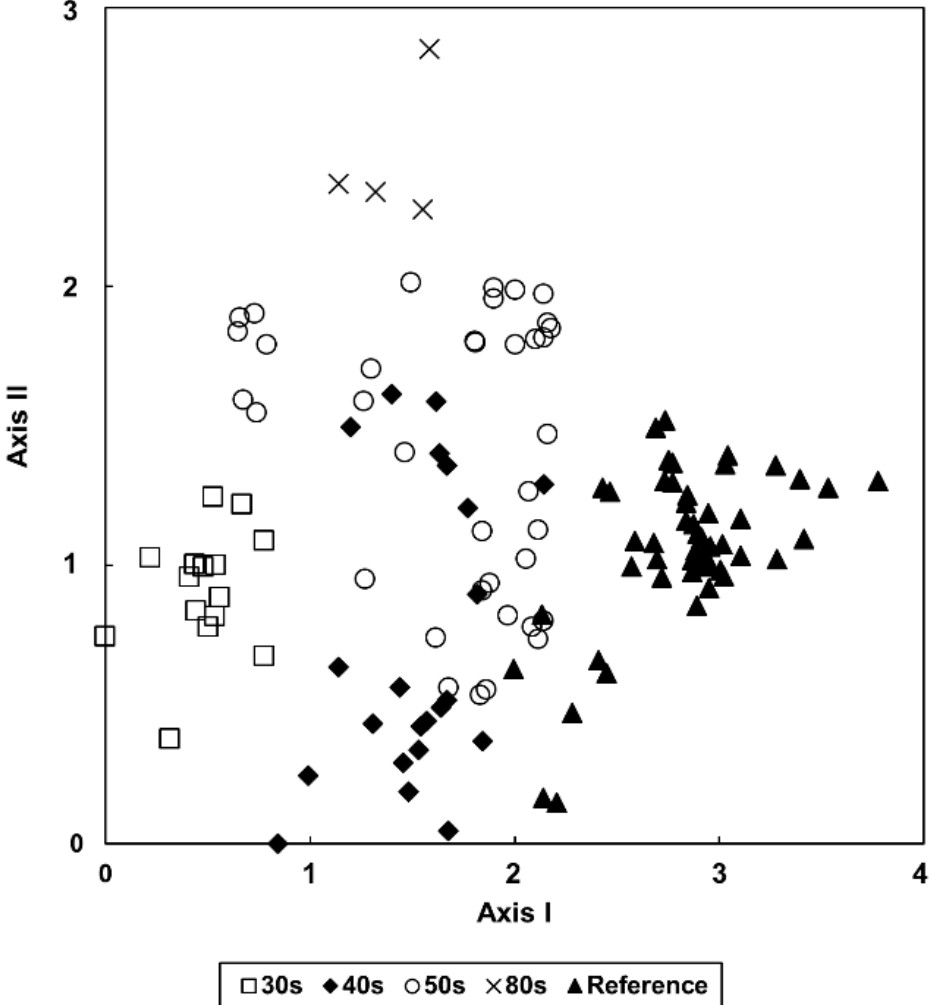

**Figure 4.** Ordination of pitch pine stands with different ages and oak stands selected as the reference stand. 30s, 40s, 50s, and 80s indicate 30-, 40-, 50-, and 80-year-old stands. Reference means oak forest selected for comparison.

### 3.2. Species Diversity

Rank–abundance curves of pitch pine plantations with different ages showed usually diverse species composition and stable feature particularly in old stands (Figure 5). Species richness in the 30-, 40-, 50-, and 80-year-old pitch pine stands had values of 35, 58, 53, and 92, respectively, and that of the reference oak was 85. Shannon-Wiener's diversity indices for these stands were 1.72, 2.41, 2.27, and 2.96 in the order of stand age and that of the reference oak was 2.99. Even the youngest stand had a diverse set of species, with more than 30 species per 1600m$^2$; and the oldest one had as many as 90 species in the same areal size (Figure 5). Rank–abundance relationships (Figure 5) revealed two trends of net increases in species diversity. First, species richness showed an increasing trend in response to stand age. Second, the degree of dominance, determined by the steepness of the curves, declined in response to stand age. In response to stand age, the relative abundance of average species became higher, indicating a more even distribution of space occupancy. In consequence, both richness and evenness increased over years after afforestation.

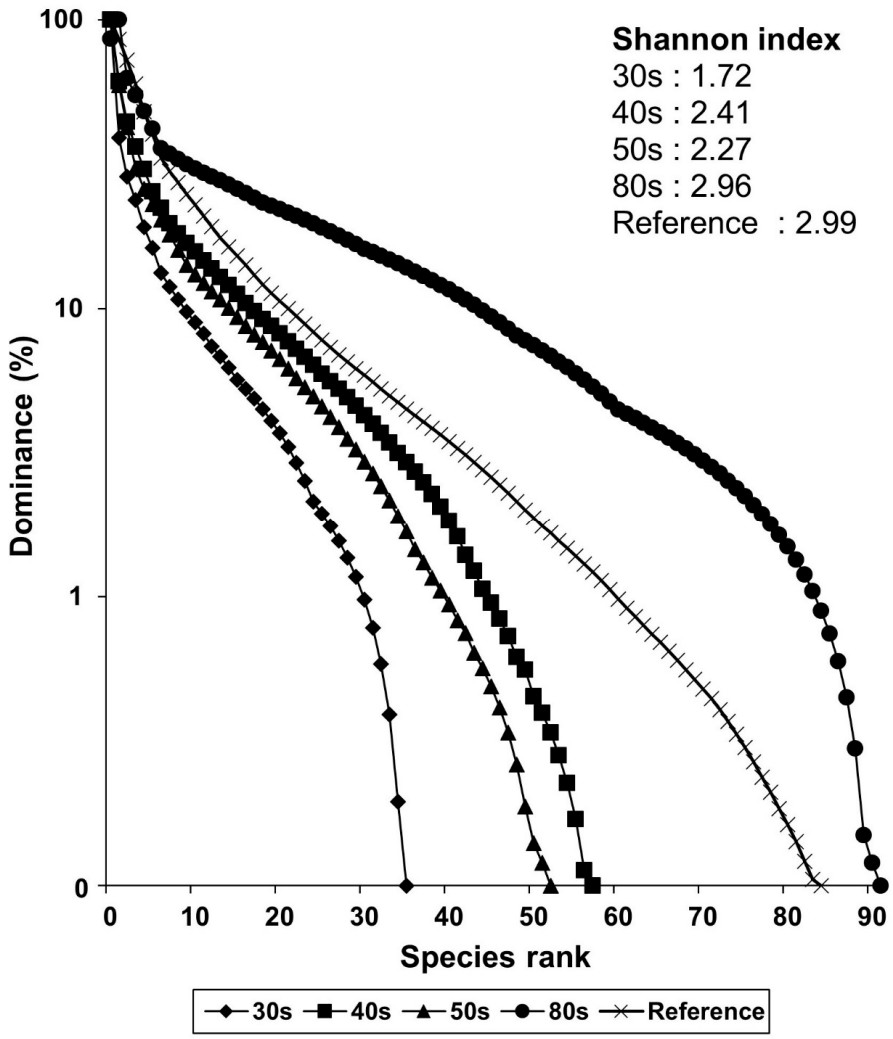

**Figure 5.** Rank–abundance curves of pitch pine stands with different ages and oak stands selected as the reference stand. 30s, 40s, 50s, and 80s indicate 30-, 40-, 50-, and 80-year-old stands. Reference means oak forest selected for comparison.

*3.3. Canopy Profile*

The canopy profile of 30-year-old pitch plantations lacked an understory tree layer and the coverage of the shrub and herb layers was very low. The 40-, 50-, and 80-year-old pitch pine plantations had a canopy profile of four layers including canopy tree, understory tree, shrub, and herb layers, and the coverage of shrub and herb layers also increased. Compared to that of the reference oak forest, coverage of the understory tree layer in the 40-, 50-, and 80-year-old pitch pine plantations was higher, whereas those of the shrub and herb layers were lower. In the older plantations, the coverage of the understory tree layer, composed of trees that become replacer species, sometimes became similar to or surpassed that of the pitch pine forming canopy tree layer. In the canopy profile of the reference oak forest, the coverage of canopy tree, understory tree, shrub, and herb layers cover the ground at values of 80%, 25%, 40%, and 50%, respectively (Figure 6).

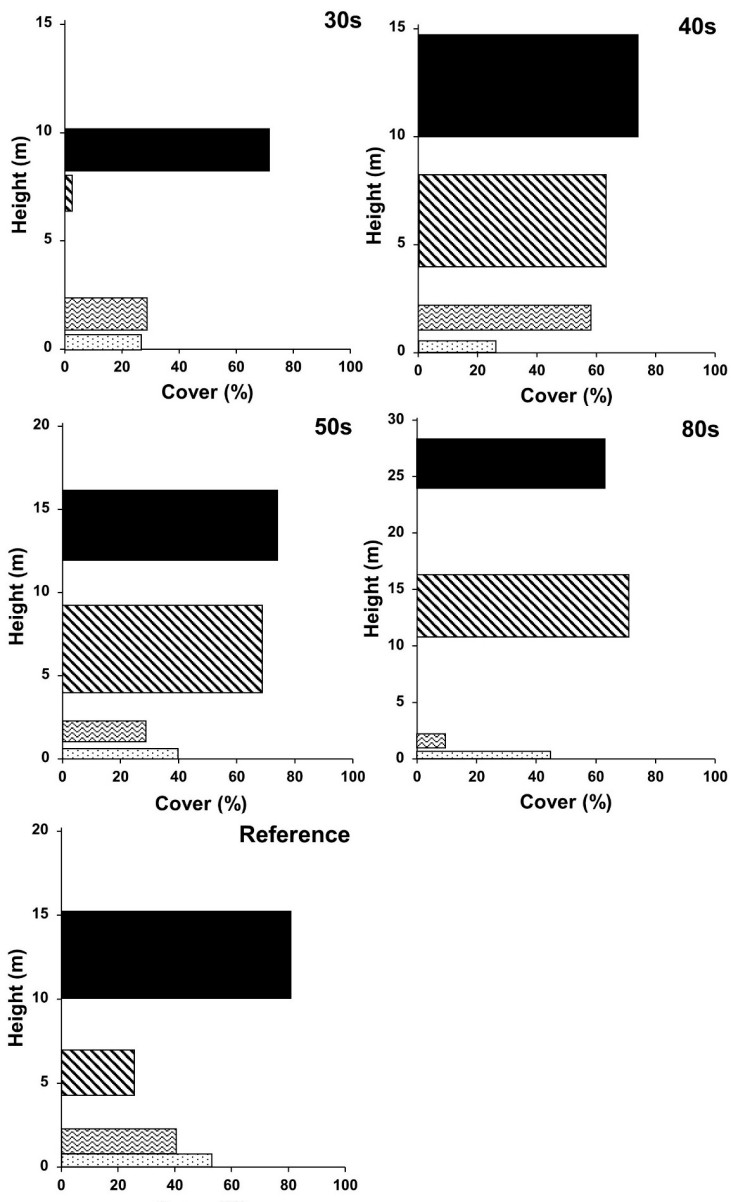

**Figure 6.** Canopy profiles of pitch pine stands with different ages. The thickness of the horizontal bars represents the span of the canopy height, and the length of each bar represents the total cover of that canopy. 30s, 40s, 50s, and 80s indicate 30-, 40-, 50-, and 80-year-old stands.

*3.4. Vegetation Dynamics*

In the diagram expressing the frequency distribution by diameter class of the dominant tree species in 30-year-old pitch pine plantations, pitch pines showed a normal distribution pattern in the diameter classes from 15.1 cm to 24.0 cm, whereas oaks showed a reverse J-shaped pattern in the diameter classes below 6.0 cm. In the diagrams of pitch pine plantations of the older stand ages, diameter class of both the pitch pine and oak increased but showed the same pattern as that of 30-year-old plantation. In each diagram, other species showed a pattern similar to that of the oak, but the frequency was far lower than that of the oak. In a diagram of the reference oak forest, Mongolian oak appeared in all the diameter classes from 0 to 40 cm. The distribution of diameter classes was divided into two groups: mature trees and juvenile trees. The former group showed a normal distribution in a range from 4.1 to 40.0 cm, with peaks in the 16.1 to 28.0 cm range. The juvenile tree group was restricted to below the 4.0 cm class (Figure 7).

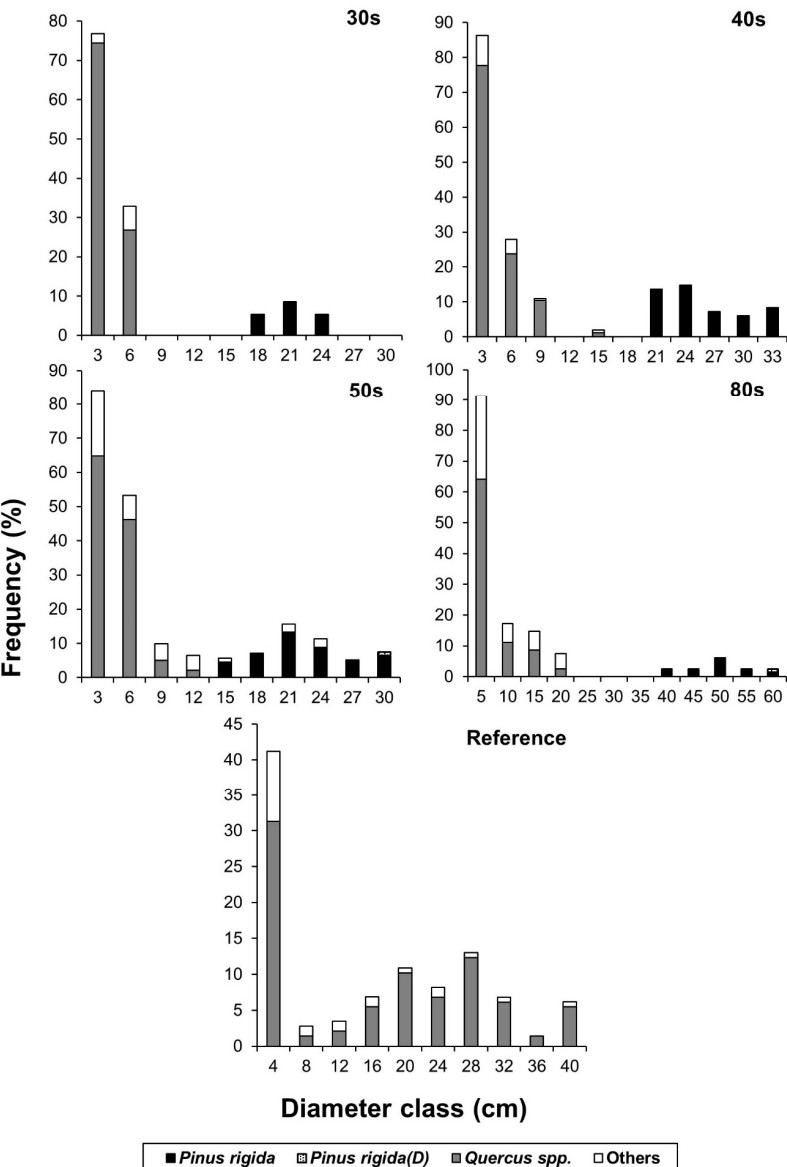

**Figure 7.** Diameter class distribution diagram of the tree species in pitch pine stands with different ages. 30s, 40s, 50s, and 80s indicate 30-, 40-, 50-, and 80-year-old stands. Reference means oak forest selected for comparison. D indicates dead tree.

*3.5. Physic-Chemical Property of Soil*

The thickness of litter layer was the thinnest in 30-year-old pitch pine plantation as 2.11 cm and tended to increase with stand age, reaching 3.40 cm in 80-year-old plantations. In the reference oak forest, this value was a little higher than in the pitch pine plantations as 4.00 cm. pH also tended to increase with stand age from 4.24 to 4.98 and that of the reference oak forest was higher than those in the pitch pine plantations as 5.48. Organic matter content usually showed an increasing trend with stand age, but the values fluctuated from 4.31% in 40-year-old pitch pine plantations to 7.88% in 50-year-old plantation depending on stand age. The value of the reference oak forest was a little higher than in pitch pine plantation as 8.49%. Total nitrogen content was the lowest in 30-year-old pitch pine plantation as 0.03% and tended to increase with stand ages, reaches to 0.13% in the 80-year-old plantation. The value of the reference oak forest was a little higher than in pitch pine plantation as 0.16%. Available phosphorus content was the lowest in 30-year-old pitch pine plantation as 2.5 ppm and tended to increase with stand age, reaching 35.0 ppm in the 80-year-old plantations. In the reference

oak forest, this was a little higher than in pitch pine plantations as 43.0 ppm. Exchangeable potassium ($K^+$) content fluctuated from 0.27 to 0.45 cmol+/kg depending on the study site, and in the reference oak forest, it was a little lower than in the pitch pine plantations as 0.23 cmol+/kg. Exchangeable calcium ($Ca^{2+}$) content was the lowest in 30-year-old pitch pine plantation as 0.68 cmol+/kg and tended to increase with stand age, reaching 1.57 cmol+/kg in the 80-year-old plantation. In the reference oak forest, it was higher than in the pitch pine plantations as 2.44 cmol+/kg. Exchangeable magnesium ($Mg^{2+}$) content did not show a big difference among pitch pine plantations with different ages, but the content changed from 0.56 cmol+/kg in the 50-year-old pitch pine plantations to 0.68 cmol+/kg in the 80-year-old plantations; in the reference oak forest, this value was a little higher than in the pitch pine plantations as 1.00 cmol+/kg (Figure 8).

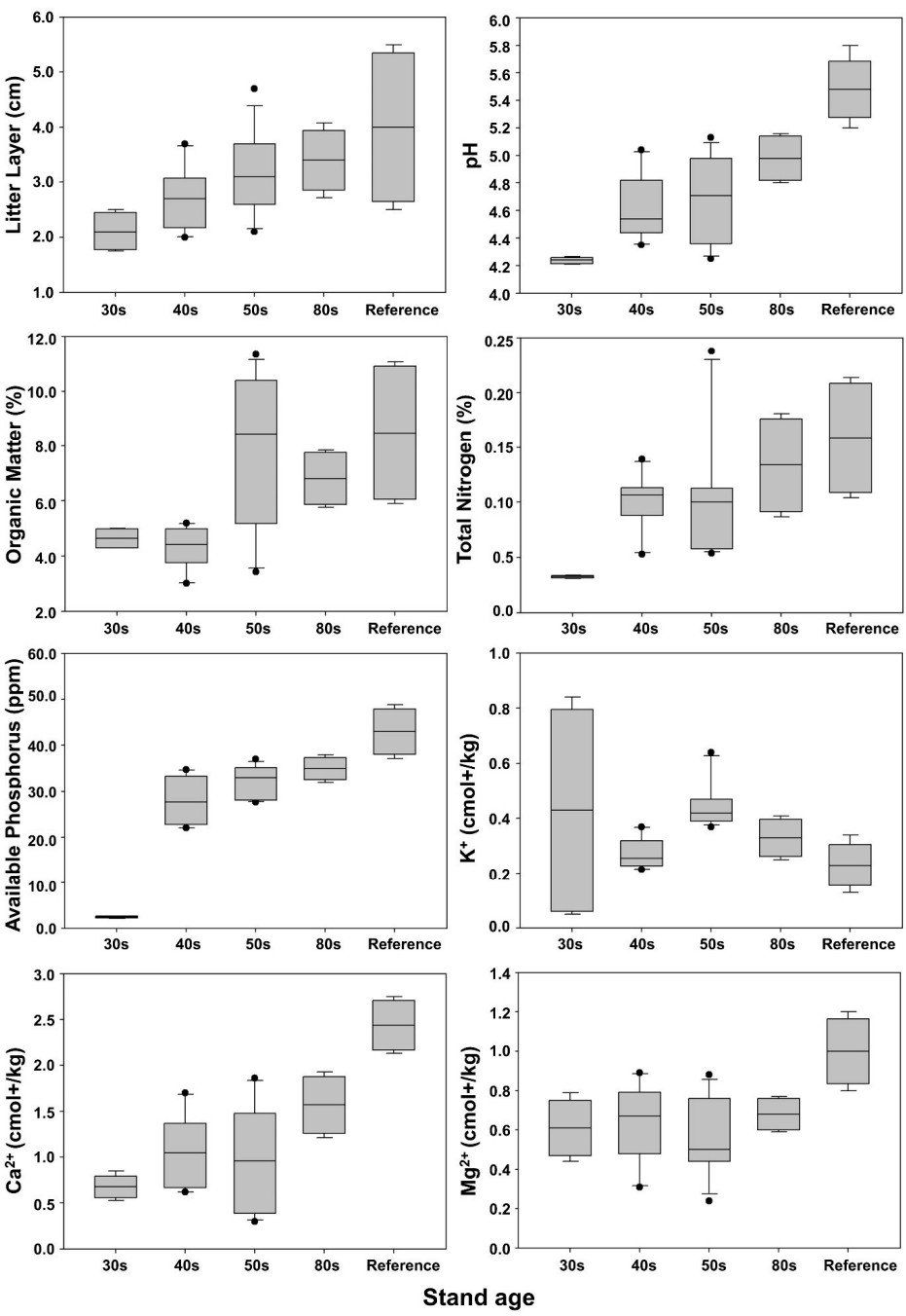

**Figure 8.** A comparison of the physic-chemical properties of the soil among pitch pine stands with different ages and reference oak stand.

The physic-chemical properties of soil in each pitch pine plantations with different age were compared to those of the reference oak forest (Figure 8). The 30-year-old pitch pine plantation showed a big difference in all soil environmental factors except $K^+$ content with the reference oak forest. The 40-year-old pitch pine plantations showed a big difference in pH, organic matter, available phosphorus, $Ca^{2+}$, and $Mg^{2+}$ contents. The 50-year-old pitch pine plantations showed a big difference in pH, available phosphorus, $K^+$, $Ca^{2+}$, and $Mg^{2+}$ contents. The 80-year-old pitch pine plantation showed a big difference in pH, available phosphorus, $Ca^{2+}$, and $Mg^{2+}$ contents with the reference oak forest.

### 3.6. Naturalization and Invasive Potential of Pitch Pine

The stand profile prepared by installing a belt transect from a mature (47-year-old) pitch pine stand to an incised slope along an expressway in Cheongyang showed that the mature pitch pine stand reproduced a new pitch pine stand by self-seeding (Figure 9). The stand profile shows that overstory of the mature stand is covered with pitch pine but native plants including oak species dominate the understory. In a case of the young (nine-year-old) stand, pitch pine dominates the stand but stratification was not yet completed (Figure 9).

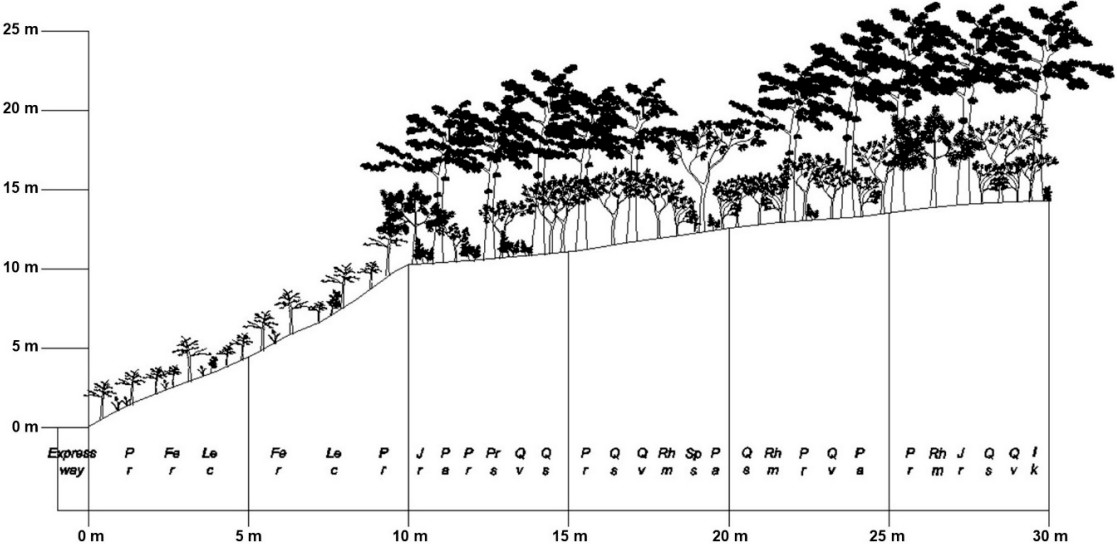

**Figure 9.** Stand profile depicted a mature pitch pine stand and a young pitch pine stand, which self-seeded from the mature stand. Fer: *Festuca rubra*, Ik: *Indigofera kirilowii*, Jr: *Juniperus rigida*, Lec: *Lespedeza cyrtobotrya*, Pa: *Pteridium aquilinum*, Pr: *Pinus rigida*, Prs: *Prunus serrulata*, Qs: *Quercus serrata*, Qv: *Q. variabilis*, Rhm: *Rhododendron mucronulatum*, Sps: *Spodiopogon sibiricus*.

In a diameter class distribution histogram of dominant tree species in the nine-year-old pitch pine stand, which self-seeded on an incised slope along an expressway, the frequency of the pitch pine was the highest frequency in the diameter class from 1.1 to 3.0 cm and decreased continuously with the increase in diameter class. Therefore, the pattern showed a reverse J-shaped pattern except in the diameter class below 1.0 cm. The other species appeared at a very low frequency (Figure 10).

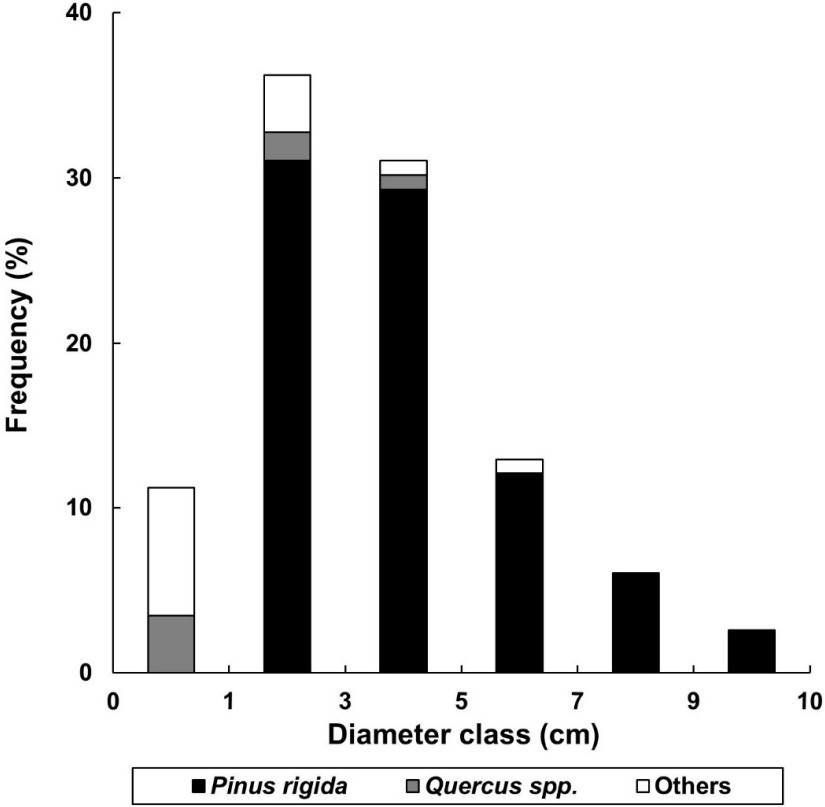

**Figure 10.** Diameter class distribution diagram of tree species appeared in a young pitch pine stand on the incised slope bordered on an expressway, which was reproduced a mature pitch pine plantation.

## 4. Discussion

### 4.1. Successional Changes of Pitch Pine Plantation

On the basis of the traditional succession theory [81,82], species composition changes as the result of modification of the abiotic environment by the community. Changes of the species composition occurs because populations tend to modify the abiotic environment, making conditions favorable for other populations until an equilibrium between biotic and abiotic factors is achieved. In the result of this study, physic-chemical properties of the soil changed significantly with plantation ages and thus reached similar level to those of the reference oak forest in 50- and 80-year old plantations (Figure 8). We evaluated changes in species composition by comparing species composition of pitch pine plantations with different stand ages with that of the native Mongolian oak (*Quercus mongolica*) community, which is the most widely distributed and representative of the late successional communities in Korea [83]. As the result of stand ordination based on vegetation data (Figure 4), the species composition of pitch pine plantations did not show remarkable changes in the years after afforestation. However, the difference in species composition between plantations and Mongolian oak stands was not as large as the difference between plantations of different in stand ages. This result implies that the species composition of pitch pine plantations is not greatly different from that of the Mongolian oak community.

Tree size distributions are useful indicators of the structure and dynamics of tree populations [84–86].

Therefore, ecologists and foresters who are concerned with species coexistence, competition, and forest management have long been interested in them [87]. The diameter class distribution of plant populations is generally depicted as frequency histograms [88]. The distribution patterns of frequency in each diameter class indicate the potential change of the population structure in the plant community. A plant population where there are numerous young individuals and fewer mature ones, is recognized as having a reverse J-shaped diameter distribution pattern [89,90]. The normal population pattern

with comparatively fewer juveniles relative to adults is typically replaced by another pattern in the future [13,74,91–93].

In our study, pitch pine occupied relatively larger diameter classes and showed a normal distribution pattern, whereas *Quercus* genus plants had relatively smaller diameter classes and showed a reverse J-shaped distribution pattern (Figure 7). We can forecast the succession of pitch pine plantations into native plant communities dominated by oaks from this result.

Most of the physic-chemical properties of the soil showed a statistically significant increase according to plantation ages and thereby became similar to those of the reference oak forest (Figure 8). Such a change led to changes species composition (Figure 4) and increases in species diversity (Figure 5) in pitch pine plantations. This result suggests that the vegetation change occurred in pitch pine plantations follows the facilitation model of succession [94]. Meanwhile, oak as a successor species tolerates shading under the pitch pine canopy. This result suggests that the successional change occurred in pitch pine plantations also follows the tolerance model [94].

### 4.2. Restoration Effects Confirmed from Pitch Pine Plantation

One way of setting a baseline and measuring restoration success is to define the normal 'biological integrity' of a system and then measure deviations from it. Integrity implies an unimpaired condition, or the quality or state of being complete or undivided [43,93].

Biological integrity is defined as "the ability to support and maintain a balanced, integrated, adaptive biological system having the full range of elements and processes expected in the natural habitat of a region" [43,95–97]. In order to evaluate a restored system, ecological attributes of the system are compared with those of the 'undisturbed' system. In our study, we compared the species composition and biodiversity of the afforested pitch pine stands with Mongolian oak stands selected as the reference stand. The species composition of pitch pine stands resembled those of the reference stands (Figure 5) as their diversity increased over time (Figure 6). In consequence, the afforestation by introducing pitch pine helped increase both biological integrity and ecological stability and thereby met the restoration goal [38,43,95–97].

On the basis of the criterion of Aronson et al. [38], reforestation carried out by introducing exotic plant species like pitch pine in South Korea corresponds to reallocation or replacement rather than restoration. However, as was shown in the results of this study, although pitch pine is an alien species, it can succeed to the native vegetation. In this respect, pitch pine plantations meet the restoration goal successfully [38]. The reforestation projects achieved successful greening, and furthermore, also had the effect of ecological restoration by recovering native vegetation and by increasing biodiversity. Owing to these results, the reforestation programs that were led by the South Korean government during the past five decades are often cited as an 'exemplary model' of success around the world [98].

As shown in Figure 2; Figure 3, pitch pine plantations were usually introduced in mountainous lowlands where the forest was devastated as the result of excessive use. Establishment and development of pitch pine plantations contributed to erosion control and thus led to development of the soil [14]. The supply of organic matter by established vegetation appeared in the increases of thickness of litter layer and organic matter content, and their decomposition led to an increase in nutrients content (Figure 8). In particular, it is believed that litters of the native broadleaved trees including oaks established and grown rapidly on the forest floor of pitch pine plantation (Figures 6 and 7) were decomposed rapidly and thus played an important role as the stand age of the pitch pine plantation increases [99–103]. Consequently, the younger stand age, the more soil environmental factors showed a big difference with the reference oak stands, and the number of environmental factors that showed a big difference tended to decrease according to the stand age of the pitch pine plantation.

Further, such an improvement in the environmental conditions caused vegetation change similarly to natural vegetation (Figure 4) and increased species diversity (Figure 5). In these serial changes, both devastated forest and extensive afforestation induced fragmentation of the natural landscape. In this respect, it can be interpreted that the succession of pitch pine plantation into natural forest through

establishment and development also contributed to recovering connectivity among natural vegetation patches and thereby improving the landscape quality [104,105].

*4.3. Naturalization and Invasive Potential of the Pitch Pine in South Korea*

Pitch pines planted in South Korea became a common tree in Korean vegetation together with native plants. Figure 9 demonstrates the typical feature of pitch pine forests in South Korea during this time. Pitch pines dominate the canopy layer and native oak (*Quercus*), azalea (*Rhododendron*), cherry tree (*Prunus*), juniper (*Juniperus*), etc., compose the understory layers.

According to Hallett's definition, naturalized plants are plants in balance with the biota of the ecosystem in which they inhabit [106]. Naturalized plants are those aliens that form self-replacing populations for at least 10 years without direct artificial intervention (or despite human intervention) by recruitment from seeds or ramets capable of independent growth based on the work of Richardson and Pyšek [53]. The vegetation that pitch pine inhabits along with native plants of Korea, and young pitch pine stands reproduced from mature pitch plantation (Figure 9) show that the pitch pine is naturalized in Korea. This result is in line with Hallett's definition for the naturalized plant [106] and Richardson and Pyšek's [53] concept of naturalization.

On the other hand, after the pitch pine afforestation, there were significant changes in soil properties, such as the thickness of the litter layer, pH, and nutrient content including total nitrogen and available phosphorus over stand age since reforestation and became similar to the soil condition of the reference oak forest (Figure 8) [14]. This result implies that pitch pine afforestation for reforesting degraded forests in South Korea was successful. The species composition of the pitch pine plantations was different depending on the study site, but became similar to that of the native oak stands chosen as the reference plots (Figure 4).

In older plantations, there was a trend for pitch pine stands to be replaced by native oak stands (Figures 6, 7 and 9).

The pitch pine stands allow succession into native oak stands (Figure 7), like native pine stands in North America do [107]. As the pitch pine trees grow, they hold the soils and prevent erosion, as well as keeping nutrients in soil [14]. Thus, pitch pines provide the environment that is suitable for native oaks to germinate and grow (Figures 6, 7 and 9) [13,14].

As the nutrient contents in the forest soil tended to increase according to stand age (Figure 8), the growth rate of the pitch pine declined, whereas that of the native oak increased [108]. In consequence, the oak stands replace pitch pine stands following the facilitation model of succession [94].

According to these results, pitch pine can be recognized as a naturalized plant, rather than an invasive plant in South Korea [53].

## 5. Conclusions

Pitch pine was typically planted in the western area of South Korea, which features a lower elevation and gentler slope than in the eastern area. The forestation was concentrated below 300 m above sea level and at slopes below 20°. On the basis of the distributional trend, it was estimated that pitch pine forestation was directed at land degraded by excessive land use in Korea [13,15].

Pitch pine forestation can be sustained without special treatment once it is established. The pitch pine can grow well in dry and infertile soil [109,110], and therefore it was appropriate for the reforestation project in the degraded forests of South Korea. The pitch pines were well established, and facilitated recovery of the species composition of the native vegetation. Consequently, pitch pine plantation secured similar species assemblage to the native vegetation and increased species diversity by facilitating indigenous species. Further, succession continues into the later stage. Soil environmental conditions become fertile in the years after forestation and thus reach a similar level to those of the reference oak stands [14,108,111]. In this respect, it can be concluded that the afforestation project of introducing pitch pine, which was started as reallocation, the lowest level of restoration [38], helped to achieve the restoration goal [44,96,97,112].

Introducing alien species may be considered as a method to restore the environment when it has been seriously degraded. However, we should be aware that the possible unexpected impacts of invasion are considered to be among the major threats to biodiversity [113]. A lot of invasive species were introduced accidentally but many pine invasions started from extensive plantation [17,114]. In Korea, the pitch pine was purposefully introduced and is now well established and continues to reproduce by themselves without human support in the disturbed habitats. Some young pines occasionally spread out from pitch pine plantations to the edge of the forestation or to open soil near a road. However, the species did not become invasive [53]. The assumed cause is that the competitive native species suppressed the pitch pine, a shade intolerant under the integrate vegetation structure and the rich soil conditions that the successful restoration led to.

**Author Contributions:** Writing - original draft, H.L.; Investigation, Data curation and Formal analysis, H.C.S.; Investigation, Data curation and Formal analysis, J.H.A.; Conceptualization, Supervision, Writing - review & editing, C.S.L. All authors have read and agreed to the published version of the manuscript.

**Funding:** This research received no external funding.

**Conflicts of Interest:** There are no potential conflicts of interest to declare.

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
