# Peer review of "Assessment of Restoration Effects and Invasive Potential Based on Vegetation Dynamics of Pitch Pine (Pinus rigida Mill.) Plantation in Korea"

_forests, doi:10.3390/f11050568_

Round 1

Reviewer 1 Report

Overall, the written English needs improvement. There is too much use of inappropriate or incorrect terminology, grammar and style errors, etc.

Introduction:

Ln 51 – Scientific name of a species needs to include the authority.

Ln 51 – The 1900s is the 20th century.

Ln 95-98 – I’m not clear on how objectives 1 and 3 differ.

Methods:

Ln 105-107 – How can you be sure that the reference sites are/were suitably similar to the pitch pine plantation sites to serve as references? What was the reason these sites were not planted and allowed to regenerate naturally? Are the native oak stands the same age as the pine plantations? You haven’t provided nearly enough information about the oak stands so that the reader can evaluate their suitability as references.

Section 2.2 – I’m not sure what the purpose of this section is. Why are these maps being constructed? I suspect that this section needs a little bit more explanatory detail.

Section 2.3 – I’m not sure you’ve adequately explained your vegetation analyses, although some of this may be a writing problem. Were species and ground cover the only measurements taken? You might brief explain the Braun-Blanquet approach. How was ground cover measured over a 20 m X 20 m plot? Was importance value simply based on the ground coverage of a species relative to other species? That is not the typical way of assessing importance value in the ecological literature. Maybe you could provide the formula used to calculate importance value.

Ln 161-162 – Rank-abundance curves provide species richness, but to not provide a direct measure of diversity.

Ln 163-165 – How were ‘layers’ of the canopy delineated? How does coverage contribute to the canopy profile? This section needs more detail.

Ln 166-169 – What were the “major tree species”? What percentage of the total species composition did this entail? If you were measuring stem dimeters, why didn’t you use basal area to determine species dominance, which is the more common approach? How was the diameter class distribution used to predict successional trends, particularly if not all species were included?

Section 2.5 – Needs more detail.

Ln 186-187 – You don’t have pretreatment soil analysis on any of your sites, so you don’t know what differences existed before the planting of pine. How, then, can you attribute differences in soil factors to stand age?

Overall, I think that the Methods section lacks detail for the reader to truly understand exactly what was done and how in this study.

Results:

I’m not a user of DCA, so I don’t claim to fully understand its usage. However, when I see the results, such as presented in Figure 4, they generally explain what factors to into each axis.

Ln 205-213 – Again, rank-abundance curves show species richness, but not necessarily species diversity. Although, as you point out, they can give you an idea of evenness. Why didn’t you calculate a Shannon’s diversity index? It seems as though you have to data to do so.

Figures 6 & 7 – You’re showing two panels in each figure for stands in the 40s, and three panels for stands in the 50s. You don’t make this distinction in figure 5. Obviously, you’re showing one panel for each of your seven sites. Why haven’t you combined the sites of the same age class as you did in figure 5?

Table 2 gives the significance probably of each factor, but does not tell you which stand ages differ from which. Figure 8 suffers from the same problem. There needs to be an indication of which ages are statistically different from which other ages.

Figures 9 and 10 are nice anecdotal descriptions, but without replication I’m not sure how much it adds to your analysis. It simply shows that on a single, highly disturbed site, that pitch pine was able to reproduce from seed from an adjacent mature pine stand.

Discussion:

Section 4.1 – This section is largely redundant, and the parts that are not probably belong in the Introduction, not the Discussion.

Section 4.2 – Your analysis of successional changes is based strictly on numbers of species and the relative dominance of the species. Nowhere do you show or explain whether the species found in the pine plantations are the same species found in the oak stands. Getting a diverse set of species back into the system is good, but if they are not the same set of species as occurs in the native communities, then you haven’t restored the native vegetation.

Section 4.4 – Your study has provided no evidence that pitch pine is widely naturalized in Korea. A single example of pine regeneration on a severely disturbed site adjacent to a mature pine stand is not adequate to make blanket statements about pitch pine naturalization.

Ln 378 – Please note that science generally does not “prove” things. Rather, a study may support a theory of concept. In this case, a single example certainly doesn’t even provide much support for the idea that pitch pine is naturalized in Korea. That doesn’t mean that I isn’t, but only that your study does not support your statement.

Ln 379-381 – Most of the trends in your soil analyses are logical and predictable following afforestation of degraded sites, and the transition from pure pine to a pine-hardwood mix. I think your paper would be stronger if you explained/interpreted these trends rather than just pointing them out with a set of figures.

Ln 383-392 – Clearly the pine stands are succeeding on to pine-hardwood stands, and will likely eventually become hardwood stands. It would be nice to see some number of stems per hectare illustrating the mortality in the pine component that is allowing for the replacement by hardwoods.

Conclusions:

Ln 399-405 – This is not a conclusion. It is simply the restatement of material that has already been stated once or twice before in the manuscript.

Author Response

I revised my manuscript by referring to reviewer's comments.

I am uploading my revised editions (response to reviewer's comment, revised manuscript (with track and clean).

Reviewer 2 Report

My comments (forests-736030):

This manuscript, entitled “Assessment of restoration effects and invasive potential based on

vegetation dynamics of Pitch pine (Pinus rigida Mill.) plantation in South Korea”, was intended to examine the restoration effects from afforested pitch pine plantations in denuded areas in South Korea for the past years and further assess the exotic potential of the exotic pitch pine species in the nation. The results showed that species diversity increased, and physic-chemical properties of soil were improved in the pitch pine plantations after afforestation. The frequency distribution of diameter classes of major tree species and the changes in canopy profiles in the different aged stands of pitch pine plantations showed a successional trend to native oak stands. Thus, the authors conclude that such changes in plant species diversity and soil properties in the pitch pine plantation satisfy the restoration goal successfully. Pitch pine forests were successfully established in Korea and thereby the species has been naturalized but not become invasive.

Afforestation is an important topic because we need more forest resources and forest ecosystem services, especially under the global change. The results from this study provided us useful data and information for further understanding of invasive tree species in specific region. The contents of this manuscript fit the scope of ‘Forests’ for publication well. However, it seems to me that there are some critical shortcomings in this manuscript, which prevent this manuscript from publication. There is no ‘Experimental design’ in the section of ‘Materials and Methods’. The information for describing the studied sites is not provided adequately. The basic abiotic conditions of the selected sites are not comparable. The preparation of the manuscript is not professional. Therefore, I recommend this manuscript be rejected, but can be re-submitted after providing a clear description of the experimental design and provide more data and information. Here. I would like to provide my suggestions and commends in detail below for consideration when the authors revise the manuscript.

Line 101 (L101), in the ‘Study areas’, more information about environmental factors in each selected site should be provided, such as annual mean precipitation, temperature, soil type etc. Because the locations of the selected 7 sites cover a large distance from south to the north of the country (Fig.1), the authors should provide studied sites’ information enough. In addition, more information about pitch pine plantations in the 7 sites should be provided, such as mean DBH, mean tree height, stand density, major plant species.

The 7 selected sites represent different aged stands of pitch pine plantations (33, 46, 48, 52, 53, 56, and 86 years), do the ‘native oak stands, which are located near to each study sites’ represent the corresponding years?  How old of the 49 oak stands are?

L133-136, the data and information in the Figure 3 are confused. It seems that there are 12 sites in the top sub-figure, 10 sites in the middle sub-figure and 8 sites in the bottom sub-figure. How many sites are there in this study?

L137, after ‘Study areas’, the authors should provide ‘Experimental design’.

L149, there are 15, 13, 7, 15, 8, 16, and 4 plots for the 7 selected sites pitch pine plantations, how far away is for each plot in a site? Do the characters of pitch pine stands in a site are similar? Because this study employed an approach of chronosequences, meaning using various-spatial sites to replace various-temporal sites, it is critical to ensure other conditions of all selected stands should be similar, besides the age. But this ‘spatial replace temporal’ approach assumes that a set of differently aged plantations were similar with respect to other state factors such as climate conditions, soil type, parent material and topography. Otherwise, this approach will introduce great uncertainties. Are the stand densities of the selected stands similar? Are the soil types for all stands similar? Do the management practices for all pitch pine plantations Similar?

L166, the values of DBH for the selected pitch pine plantations were presented in the manuscript (Figure 7), but I did not see the ‘stem diameters at stem base for seedlings and saplings’ in the manuscript. How old of the tree were seedlings and saplings in this study?

L167, what are the names of the major tree species? The authors should list the names of several major tree species, saying the names of the top five major tree species (based on the importance values) in each site.

L201, what are the ages of the reference oak forests in the Figure 4? If the authors want to make a comparison, the similar aged stands of pitch pine plantations and oak forests should be used. That is, 30s-year-old pitch pine plantations compared with 30s-year-old oak forests, … and 80s-year-old pitch pine plantations compared with 80s-year-old oak forests in this study. In addition,  the quantitative data should be presented when the authors describe the results from Figure 4.

L205, How did the authors develop the rank-abundance curves? If the corves were developed based on the selected stands for each age class, it would introduce uncertainties because the total areas of species survey were different for each aged site in this study. What is the minimum area for the vegetation survey in the study area? The 30s-, 40s-, 50s-, 80s-year-old pitch pine plantations, and oak forests had 15, 20, 39, 4, and 49 plots, respectively. The corresponding areas for these age-class stands were 6000, 8000, 15600, 1600 and 19600 m2.

L208, it should be (Figure 5).

L208-213, the sentences should be combined into the ‘Materials and Methods’ section or ‘Discussion’ section.

L211, what are ‘average species’?

L214-217, I would like to see ‘quantitative data’, not ‘quality data’ when to describe the results in Figure 6.

L218-220, Control or Reference? Choose only one and keep consistent.

L223, for Figure 6.  It would be better if only one canopy profiles of the 30s-, 40s-, 50s-, 80s-year-old pitch pine plantations and oak forests was provided, instead of one for 30s-, two for 40s-, three for 50s-, one for 80s- year-old pitch pine plantations and 0 for oak forests. Additionally, it would be better if using different colors or textures of the horizontal bars to represent canopy tree, understory tree, shrub and herb layers.

L226-229, I would like to see ‘quantitative data’, not ‘quality data’ when to describe the results in Figure 7.

L231, for Figure 7, keep the vertical axes the similar for all sub-figures in Figure 7, such as 0, 20, 40, 60, 80. Not some are 0, 20. 40, 60; some are 0, 10, 20, 30, 40, 50. It would be better if only one of the selected age-class stands was presented in this figure.

L235-237, it would be better if ‘quantitative data’, not ‘quality data’ were used to describe the results in Figure 8 and Table 2. I am surprised that why the authors liked to use the quality data in the manuscript. Science needs quantitative data.

L252, how old is a mature pitch pine stand? Is it a 30s-, 40s-, 50s-, or 80s-year-old stand?

L252-256, need quantitative data for description.

L263-265, need quantitative data for description.

L272-296, this section (4.1.) is not discussion and it should be combined into the ‘Introduction’ section.

L299-303, the sentences are confused and need to re-write.

L311-312, ‘This result implies that species composition of pitch pine plantation is not so large different from…’. What is a large different? Besides considering the total number of the plant species appeared in the different age-class stands, also need to consider how many same species were showed up in both aged-class stands and how many species were only showed up in one age-class stand.

L369, what is the meaning of ‘became one of the trees?

Author Response

I revised my manuscript by referring to reviewer's comments.

Reviewer 3 Report

This paper's subject is interesting, because it looks at the succession of plantations whose objective was to halt erosion and restore forest cover, not for production. It also looks at the potential for invasiveness of an exotic species.

However, I have many questions about both, the selection of some sites (eg one site at a much higher altitude than the rest as only representative of old plantation); and the presentation of the results: I think it would be better to compare plantation sites to their reference oak stand, rather than sites in different parts of the country, with different vegetation, to a merged reference stand.

I'd also like to see a bit more about the characteristic species of the stands at different stages; or at least groups of species (shrubs, herbaceous, lower plants if surveyed).

I also think that the study about invasiveness, although interesting, should be a bit more separated from the other section, eg by not including Cheongyang in table 1 with the rest without any clarification.

I have many other suggestions in the attached PDF.

Author Response

I revised my manuscript by referring reviewer's comments.

Round 2

Reviewer 1 Report

I didn't see significant improvements over the initial submission. Despite considerable editing, the writing still needs improvement. Since you didn't provide any write-up on how you responded to my previous comments, I'm not sure what substantive changes you have made. However, it doesn't appear that you addressed several of my more important comments. The description of your methodologies still needs greater detail. You have not justified your soil analyses, particularly since you have no pre-planting soil information. Your assessment of the risk of invasiveness from pitch pine is unreplicated, and therefore lacks any real scientific substance.

Author Response

Response to reviewer’s comments (Reviewer 1)

I didn't see significant improvements over the initial submission. Despite considerable editing, the writing still needs improvement. Since you didn't provide any write-up on how you responded to my previous comments, I'm not sure what substantive changes you have made. However, it doesn't appear that you addressed several of my more important comments. The description of your methodologies still needs greater detail. You have not justified your soil analyses, particularly since you have no pre-planting soil information. Your assessment of the risk of invasiveness from pitch pine is unreplicated, and therefore lacks any real scientific substance.

☞ We revised our manuscript by accepting reviewer’s comments and provided write-up on those comments in our manuscript (1st and 2nd revised editions) and response on reviewer’s comments. We add those responses below.

Response to Reviewer 1 (1st response provided write-up on reviewer’s comments)

Overall, the written English needs improvement. There is too much use of inappropriate or incorrect terminology, grammar and style errors, etc.

Introduction:

Ln 51 – Scientific name of a species needs to include the authority.

☞ We revised our manuscript by accepting reviewer’s comment. Line 52.

Ln 51 – The 1900s is the 20th century.

☞ It was first introduced in the late 1800s and thus we expressed as 19th century.

Ln 95-98 – I’m not clear on how objectives 1 and 3 differ.

☞ We revised our manuscript by accepting reviewer’s comment. Lines 108 – 111.

Methods:

Ln 105-107 – How can you be sure that the reference sites are/were suitably similar to the pitch pine plantation sites to serve as references? What was the reason these sites were not planted and allowed to regenerate naturally? Are the native oak stands the same age as the pine plantations? You haven’t provided nearly enough information about the oak stands so that the reader can evaluate their suitability as references.

☞ We revised our manuscript by explaining this part in more detail.

Lines 117 – 120, lines 140 – 145, lines 166 - 171

Section 2.2 – I’m not sure what the purpose of this section is. Why are these maps being constructed? I suspect that this section needs a little bit more explanatory detail.

☞ We prepared this section to explain the background that we introduced pitch pine in Korea. As we explained in our manuscript, we just introduced pitch pine to prevent soil erosion occurring in the denuded forest due to the 2nd world war and Korean war as well as excessive use without any ecological consideration. Although this reforestation was started as the reclamation level, those plantations are now succeeding to forests close to the native forest in species composition and diversity of vegetation. That is, the plantation achieved forest recovery of the restoration level.  

Section 2.3 – I’m not sure you’ve adequately explained your vegetation analyses, although some of this may be a writing problem. Were species and ground cover the only measurements taken? You might brief explain the Braun-Blanquet approach. How was ground cover measured over a 20 m X 20 m plot? Was importance value simply based on the ground coverage of a species relative to other species? That is not the typical way of assessing importance value in the ecological literature. Maybe you could provide the formula used to calculate importance value.

☞ In the past, Braun-Blanquet method was focused on vegetation classification. But we didn’t apply the method strictly but we just evaluated cover degree based on Braun-Blanquet scale to investigate species composition. Importance value is usually obtained by integrating the relative cover, density, and frequency. But when we investigate including all of the tree, shrub, and herb, cover is significant to reduce a variation depending on plant size. Therefore, we regarded the relative coverage as the importance value. This method is generally accepted international journals:

Lee, C.S., You, Y.H., Robinson, G.R. 2002. Secondary succession and natural habitat restoration in abandoned rice fields of central Korea. Restoration Ecology 10: 306-314.

Lee, C.S., Cho, H.J., Yi, H. 2004. Stand dynamics of introduced black locust (Robinia pseudoacacia L.) plantation under different disturbance regimes in Korea. Forest Ecology and Management 189/1-3: 281-293.

Lee, C.S., Moon, J.S., Cho, Y.C. Cho. 2007. Effects of soil amelioration and tree planting on restoration of an air-pollution damaged forest in South Korea. Water, Air and Soil Pollution 179: 239-254.

Lee, C.S., Lee, A.N., Cho, Y.C. 2008. Restoration Planning for the Seoul Metropolitan Area, Korea. In: M.M. Carreiro et al (eds.), Ecology, Planning, and Management of Urban Forests: International Perspectives. Springer, New York. pp. 393-419.

Lim, C.H., An J.H., Jung, S.H., Lee, C.S. 2018. Allogenic succession of Korean fir (Abies koreana Wils.) forests in different climate condition. Ecological Research 33: 327-340.

Lee, C.S., Robinson, G.R., Robinson I.P., Lee, H. 2019. Regeneration of pitch pine (Pinus rigida) stands inhibited by fire suppression in Albany Pine Bush Preserve, New York. J. For. Res. DOI 10.1007/s11676-018-0644-3.

Ln 161-162 – Rank-abundance curves provide species richness, but to not provide a direct measure of diversity.

☞ Rank – abundance curve provides species richness and furthermore, the gradient provides the meaning of evenness. Diversity has the meaning that richness and evenness are combined. Therefore, we expressed so.

Furthermore, we added diversity index by accepting reviewer’s comment. Lines 277 – 288, Figure 5.

Ln 163-165 – How were ‘layers’ of the canopy delineated? How does coverage contribute to the canopy profile? This section needs more detail.

☞ We classified the canopy layer into four layers of canopy, understory, shrub, and herb layers and estimated coverage of each layer based on Domin-Krajina scale. Lines 226 – 230.

Ln 166-169 – What were the “major tree species”? What percentage of the total species composition did this entail?

☞ Major tree species indicate dominant species of plant communities, which form forest vegetation. We involved all tree species for this analysis.

If you were measuring stem dimeters, why didn’t you use basal area to determine species dominance, which is the more common approach?

☞ When we investigate species composition, basal area is used to determine species dominance. But as we mentioned above, we carried out vegetation survey including all of the tree, shrub, and herb. In this case, cover is usually used to determine species dominance, particularly to reduce a variation depending on plant size.

How was the diameter class distribution used to predict successional trends, particularly if not all species were included?

☞ This method is usually used to predict successional trends under the hypothesis that diameter is proportionated to age and dominant species accompany companion species in the successional process. In our manuscript, we analyzed the successional trend based on diameter class distribution histogram. Change of species composition was considered by applying DCA ordination.

We explained this part in detail to reduce reader’s misunderstanding in Discussion section (Lines 399 – 424).

Section 2.5 – Needs more detail.

☞ We revised our manuscript by accepting reviewer’s comment. Lines 250 – 260.

Ln 186-187 – You don’t have pretreatment soil analysis on any of your sites, so you don’t know what differences existed before the planting of pine. How, then, can you attribute differences in soil factors to stand age?

Overall, I think that the Methods section lacks detail for the reader to truly understand exactly what was done and how in this study.

☞ We revised our manuscript by accepting reviewer’s comment. We added ‘Experimental design’ section and explained sampling and analysis methods of soil in more detail to aid reader’s understanding. Lines 162 – 179, 237 – 248.

When the planting of pitch pine was at the height, Korean forest was devastated very severely and thus its rehabilitation was very urgently required. Moreover, the socio-economic situations were also not good because much time had not passed since the 2nd World War and the Korean War. Therefore, the reforestation project was carried out without any diagnostic assessment differently from today and consequently we couldn’t get any information on the soil condition before the planting of pitch pine. Considered this situation, we chose a method that vegetation and soil conditions of pitch pine plantations with different stand ages compare to those of the reference oak stands, which represents the natural forest in Korea.

Results:

I’m not a user of DCA, so I don’t claim to fully understand its usage. However, when I see the results, such as presented in Figure 4, they generally explain what factors to into each axis.

☞ You are right. By the way, first, we focused on changes of vegetation structure including species composition over stand age. Then, we addressed changes of soil factors separately. We would like to focus on changes of vegetation and soil traits in pitch pine plantations with different ages over stand age and comparison to those of the reference natural forests rather than relationship between the spatial arrangement of stands and the soil environmental factors.

Ln 205-213 – Again, rank-abundance curves show species richness, but not necessarily species diversity. Although, as you point out, they can give you an idea of evenness. Why didn’t you calculate a Shannon’s diversity index? It seems as though you have to data to do so.

☞ We revised our manuscript by accepting reviewer’s comment. Lines 277 – 288, Figure 5.

Figures 6 & 7 – You’re showing two panels in each figure for stands in the 40s, and three panels for stands in the 50s. You don’t make this distinction in figure 5. Obviously, you’re showing one panel for each of your seven sites. Why haven’t you combined the sites of the same age class as you did in figure 5?

☞ We revised our manuscript by unifying those graphs as the same type.

Table 2 gives the significance probably of each factor, but does not tell you which stand ages differ from which. Figure 8 suffers from the same problem. There needs to be an indication of which ages are statistically different from which other ages.

☞ We revised our manuscript by accepting reviewer’s comment. Figure 8, Table 3.

Figures 9 and 10 are nice anecdotal descriptions, but without replication I’m not sure how much it adds to your analysis. It simply shows that on a single, highly disturbed site, that pitch pine was able to reproduce from seed from an adjacent mature pine stand.

☞ We observed such phenomena throughout about 60 km on both sides of this expressway and the pattern was very similar throughout the whole range of the road. Therefore, we installed our study plot (10 m х 40 m) in one sector typical of the incised slope bordered on the expressway and collected data.

Discussion:

Section 4.1 – This section is largely redundant, and the parts that are not probably belong in the Introduction, not the Discussion.

☞ We revised our manuscript by moving a part of this section to ‘Introduction’ section and deleting the other part. Lines 47 – 51, 55 – 67, 398 – 400.

Section 4.2 – Your analysis of successional changes is based strictly on numbers of species and the relative dominance of the species. Nowhere do you show or explain whether the species found in the pine plantations are the same species found in the oak stands. Getting a diverse set of species back into the system is good, but if they are not the same set of species as occurs in the native communities, then you haven’t restored the native vegetation.

☞ We explained it as the follows. Species composition of the pitch pine plantations resembles that of the reference oak stands. As you know, distance among stands in the result of DCA ordination indicates similarity or dissimilarity of species composition among stands. In our result, distance between pitch pine stands and the reference oak stands was not farther than the distance among pitch pine stands of the same age stage. Therefore, we interpreted our result so. Lines 267 – 268, 410 – 415.

In addition, all plant species depicted in stand profile of 47-year-old pitch pine plantation of Figure 9 are the native plants except pitch pine of canopy tree layer. These results indicate that the pitch pine plantations restored the same set of species as occurs in the native communities. Lines 378 – 380. 472 - 473.

Section 4.4 – Your study has provided no evidence that pitch pine is widely naturalized in Korea. A single example of pine regeneration on a severely disturbed site adjacent to a mature pine stand is not adequate to make blanket statements about pitch pine naturalization.

☞ As was shown in our results, pitch pines produce seeds and they germinate successfully, giving birth to seedlings in Korea. Therefore, they can produce new stand in the disturbed areas. However, since they are early successional species, natural regeneration of the forest does not take place, and succeeded to the forest of late successional stages.

Based on previous studies, naturalized plants are those aliens that form self-replacing populations for at least 10 years without direct intervention by people (or despite human intervention) by recruitment from seeds or ramets capable of independent growth. In particular, reproduction is functioned as a significant evidence determining naturalization. Based on the criteria, we evaluated that pitch pine was naturalized in Korea. Lines 474 – 481.

In addition, as is shown in Figure 2, pitch pine plantations are established in nation-wide and the pitch pine reproduce and generate young stands centering on the disturbed land as in our study site in those areas too. So, we interpreted so.

Ln 378 – Please note that science generally does not “prove” things. Rather, a study may support a theory of concept. In this case, a single example certainly doesn’t even provide much support for the idea that pitch pine is naturalized in Korea. That doesn’t mean that I isn’t, but only that your study does not support your statement.

☞ We revised our manuscript by accepting reviewer’s comment and explained our result based on the criteria of naturalization that other ecologists suggested. Lines 470 – 481.

Ln 379-381 – Most of the trends in your soil analyses are logical and predictable following afforestation of degraded sites, and the transition from pure pine to a pine-hardwood mix. I think your paper would be stronger if you explained/interpreted these trends rather than just pointing them out with a set of figures.

☞ We revised our manuscript by accepting reviewer’s comment. Lines 461 – 466.

Ln 383-392 – Clearly the pine stands are succeeding on to pine-hardwood stands, and will likely eventually become hardwood stands. It would be nice to see some number of stems per hectare illustrating the mortality in the pine component that is allowing for the replacement by hardwoods.

☞ We revised our manuscript by accepting reviewer’s comment. Figure 7.

Conclusions:

Ln 399-405 – This is not a conclusion. It is simply the restatement of material that has already been stated once or twice before in the manuscript.

☞ We revised our manuscript by accepting reviewer’s comment. Lines 509 – 533.

Reviewer 2 Report

The quality of the revision is much improved. I am appreciated with the authors' efforts and I am also satisfied with author's responses about my comments, suggestions and questions.

Author Response

Response to reviewer’s comments (Reviewer 2)

The quality of the revision is much improved. I am appreciated with the authors’ efforts and I am also satisfied with author’s responses about my comments, suggestions and questions.

☞ Thank you very much for reviewer’s valuable comments.

Reviewer 3 Report

This version of the manuscript is significantly better than the first. Most of the points I made in the first review have been taken on board and I think the manuscript is much clearer now. Thank you!

A couple of minor points still to address: the abbreviations for the sites are different in lines 116 and 212; in particular Mt Sori is "Sb" in line 116 and "So" in line 212.

And I am not sure if this is the way the geography is described locally, but to me a site below 300 m asl is not "mountainous" (eg line 22), maybe a bit hilly...

Author Response

Response to reviewer’s comments (Reviewer 3)

This version of the manuscript is significantly better than the first. Most of the points I made in the first review have been taken on board and I think the manuscript is much clearer now. Thank you!

A couple of minor points still to address: the abbreviations for the sites are different in lines 116 and 212; in particular Mt Sori is "Sb" in line 116 and "So" in line 212.

☞ We revised it by accepting reviewer’s advice.

And I am not sure if this is the way the geography is described locally, but to me a site below 300 m asl is not "mountainous" (eg line 22), maybe a bit hilly...

☞ Our study sites are located on the lower part of a big mountain such as mountain foot or slope. Therefore, we expressed as “mountainous” rather than “hilly”.
